# Padam: Closing the Generalization Gap of Adaptive Gradient Methods in Training Deep Neural Networks

## Abstract

Adaptive gradient methods, which adopt historical gradient information to automatically adjust the learning rate, despite the nice property of fast convergence, have been observed to generalize worse than stochastic gradient descent (SGD) with momentum in training deep neural networks. This leaves how to close the generalization gap of adaptive gradient methods an open problem. In this work, we show that adaptive gradient methods such as Adam, Amsgrad, are sometimes "over adapted". We design a new algorithm, called *Partially adaptive momentum estimation method* (Padam), which unifies the Adam/Amsgrad with SGD by introducing a partial adaptive parameter $p$, to achieve the best from both worlds. Experiments on standard benchmarks show that Padam can maintain fast convergence rate as Adam/Amsgrad while generalizing as well as SGD in training deep neural networks. These results would suggest practitioners pick up adaptive gradient methods once again for faster training of deep neural networks.

## 1 Introduction

Stochastic gradient descent (SGD) is now one of the most dominant approaches for training deep neural networks (Goodfellow et al., 2016). In each iteration, SGD only performs one parameter update on a mini-batch of training examples. SGD is simple and has been proved to be efficient, especially for tasks on large datasets. In recent years, adaptive variants of SGD have emerged and shown their successes for their convenient automatic learning rate adjustment mechanism. Adagrad (Duchi et al., 2011) is probably the first along this line of research, and significantly outperforms vanilla SGD in the sparse gradient scenario. Despite the first success, Adagrad was later found to demonstrate degraded performance especially in cases where the loss function is nonconvex or the gradient is dense. Many variants of Adagrad, such as RMSprop (Hinton et al., 2012), Adam (Kingma & Ba, 2015), Adadelta (Zeiler, 2012), Nadam (Dozat, 2016), were then proposed to address these challenges by adopting exponential moving average rather than the arithmetic average used in Adagrad. This change largely mitigates the rapid decay of learning rate in Adagrad and hence makes this family of algorithms, especially Adam, particularly popular on various tasks. Recently, it has also been observed (Reddi et al., 2018) that Adam does not converge in some settings where rarely encountered large gradient information quickly dies out due to the "short momery" problem of exponential moving average. To address this issue, Amsgrad (Reddi et al., 2018) has been proposed to keep an extra "long term memory" variable to preserve the past gradient information and to correct the potential convergence issue in Adam.

On the other hand, people recently found that for largely over-parameterized neural networks, e.g., more complex modern convolutional neural network (CNN) architectures such as VGGNet (He et al., 2016), ResNet (He et al., 2016), Wide ResNet (Zagoruyko & Komodakis, 2016), DenseNet (Huang et al., 2017), training with Adam or its variants typically generalizes worse than SGD with momentum, even when the training performance is better. In particular, people found that carefully-tuned SGD, combined with proper momentum, weight decay and appropriate learning rate decay strategies, can significantly outperform adaptive gradient algorithms eventually (Wilson et al., 2017). As a result, even though adaptive gradient methods are relatively easy to tune and converge faster at the early stage, recent advances in designing neural network structures are all reporting their performances by training their models with SGD-momentum (He et al., 2016; Zagoruyko & Komodakis,

2016; Huang et al., 2017; Simonyan & Zisserman, 2014; Ren et al., 2015; Xie et al., 2017; Howard et al., 2017). Moreover, it is difficult to apply the same learning rate decay strategies that work well in SGD with momentum to adaptive gradient methods, since adaptive gradient methods usually require a much smaller base learning rate that will soon die out after several rounds of decay. We refer to it as the "small learning rate dilemma" (see more details in Section 3).

With all these observations, a natural question is:

*Can we take the best from both Adam and SGD-momentum, i.e., design an algorithm that not only enjoys the fast convergence rate as Adam, but also generalizes as well as SGD-momentum?*

In this paper, we answer this question affirmatively. We close the generalization gap of adaptive gradient methods by presenting a new algorithm, called **p**artially **ada**ptive **m**omentum estimation (Padam) method, which unifies Adam/Amsgrad with SGD-momentum to achieve the best of both worlds.

In particular, we make the following contributions:

- We propose a new algorithm, Padam, which unifies Adam/Amsgrad and SGD with momentum by a partially adaptive parameter. We show that Adam/Amsgrad can be seen as a special fully adaptive instance of Padam. The intuition behind our algorithm is that, by controlling the degree of adaptiveness, the learning rate in Padam does not need to be as small as other adaptive gradient methods to prevent the gradient update from exploding. This resolves the "small learning rate dilemma" for adaptive gradient methods and allows for faster convergence, hence closing the gap of generalization.

- We show that the Padam's performance is also theoretically guaranteed. We provide a convergence analysis of Padam in the convex setting, based on the analysis of Kingma & Ba (2015); Reddi et al. (2018), and prove a data-dependent regret bound.

- We also provide thorough experiments about our proposed Padam method on training modern deep neural architectures. We empirically show that Padam achieves the fastest convergence speed while generalizing as well as SGD with momentum. These results suggest that practitioners should pick up adaptive gradient methods once again for faster training of deep neural networks.

**Notation:**   Scalars are denoted by lower case letters, vectors by lower case bold face letters, and matrices by upper case bold face letters. For a vector $\boldsymbol{\theta} \in \mathbb{R}^d$, we denote the $\ell_2$ norm of $\boldsymbol{\theta}$ by $\|\boldsymbol{\theta}\|_2 = \sqrt{\sum_{i=1}^d \theta_i^2}$, the $\ell_\infty$ norm of $\boldsymbol{\theta}$ by $\|\boldsymbol{\theta}\|_\infty = \max_{i=1}^d |\theta_i|$. For a sequence of vectors $\{\boldsymbol{\theta}_j\}_{j=1}^t$, we denote by $\theta_{j,i}$ the $i$-th element in $\boldsymbol{\theta}_j$. We also denote $\boldsymbol{\theta}_{1:t,i} = [\theta_{1,i}, \theta_{2,i}, \ldots, \theta_{t,i}]^\top$. With slightly abuse of notation, for any two vectors $\mathbf{a}$ and $\mathbf{b}$, we denote $\mathbf{a}^2$ as the element-wise square, $\mathbf{a}^p$ as the element-wise power operation, $\mathbf{a}/\mathbf{b}$ as the element-wise division and $\max(\mathbf{a}, \mathbf{b})$ as the element-wise maximum. We denote by $\mathrm{diag}(\mathbf{a})$ a diagonal matrix with diagonal entries $a_1, \ldots, a_d$. Let $\mathcal{S}_{++}^d$ be the set of all positive definite $d \times d$ matrices. We denote by $\Pi_{\mathcal{X},\mathbf{A}}(\mathbf{b})$ the projection of $\mathbf{b}$ onto a convex set $\mathcal{X}$, i.e., $\mathrm{argmin}_{\mathbf{a} \in \mathcal{X}} \|\mathbf{A}^{1/2}(\mathbf{a} - \mathbf{b})\|_2$ for $\mathbf{b} \in \mathbb{R}^d$, $\mathbf{A} \in \mathcal{S}_{++}^d$. Given two sequences $\{a_n\}$ and $\{b_n\}$, we write $a_n = O(b_n)$ if there exists a constant $0 < C < +\infty$ such that $a_n \le C b_n$. We use notation $\widetilde{O}(\cdot)$ to hide logarithmic factors. We write $a_n = o(b_n)$ if $a_n/b_n \to 0$ as $n \to \infty$.

## 2   Review of Adaptive Gradient Methods

Various adaptive gradient methods have been proposed in order to achieve better performance on various stochastic optimization tasks. Adagrad (Duchi et al., 2011) is among the first methods with adaptive learning rate for each individual dimension, which motivates the research on adaptive gradient methods in the machine learning community. In detail, Adagrad[1] adopts the following

---

[1]The formula here is equivalent to the one from the original paper (Duchi et al., 2011) after simple manipulations.

update form:

$$\boldsymbol{\theta}_{t+1} = \boldsymbol{\theta}_t - \alpha_t \frac{\mathbf{g}_t}{\sqrt{\mathbf{v}_t}}, \text{ where } \mathbf{v}_t = \frac{1}{t} \sum_{j=1}^{t} \mathbf{g}_j^2 \qquad \text{(Adagrad)}$$

Here $\mathbf{g}_t$ stands for the stochastic gradient $\nabla f_t(\boldsymbol{\theta}_t)$, and $\alpha_t = \alpha/\sqrt{t}$ is the step size (a.k.a., learning rate). Adagrad is proved to enjoy a huge gain in terms of convergence especially in sparse gradient situations. Empirical studies also show a performance gain even for non-sparse gradient settings. RMSprop (Hinton et al., 2012) follows the idea of adaptive learning rate and it changes the arithmetic averages used for $\mathbf{v}_t$ in Adagrad to exponential moving averages. Even though RMSprop is an empirical method with no theoretical guarantee, the outstanding empirical performance of RMSprop raised people's interests in exponential moving average variants of Adagrad. Adam (Kingma & Ba, 2015)[2] is the most popular exponential moving average variant of Adagrad. It combines the idea of RMSprop and momentum acceleration, and takes the following update form:

$$\boldsymbol{\theta}_{t+1} = \boldsymbol{\theta}_t - \alpha_t \frac{\mathbf{m}_t}{\sqrt{\mathbf{v}_t}}, \text{ where } \mathbf{m}_t = \beta_1 \mathbf{m}_{t-1} + (1-\beta_1)\mathbf{g}_t, \mathbf{v}_t = \beta_2 \mathbf{v}_{t-1} + (1-\beta_2)\mathbf{g}_t^2 \quad \text{(Adam)}$$

Adam also requires $\alpha_t = \alpha/\sqrt{t}$ for the sake of convergence analysis. In practice, any decaying step size or even constant step size works well for Adam. Note that if we choose $\beta_1 = 0$, Adam basically reduces to RMSprop. Reddi et al. (2018) identified a non-convergence issue in Adam. Specifically, Adam does not collect long-term memory of past gradients and therefore the effective learning rate could be increasing in some cases. They proposed a modified algorithm namely Amsgrad. More specifically, Amsgrad adopts an additional step to ensure the decay of the effective learning rate $\alpha_t/\sqrt{\widehat{\mathbf{v}}_t}$, and its key update formula is as follows:

$$\boldsymbol{\theta}_{t+1} = \boldsymbol{\theta}_t - \alpha_t \frac{\mathbf{m}_t}{\sqrt{\widehat{\mathbf{v}}_t}}, \text{ where } \widehat{\mathbf{v}}_t = \max(\widehat{\mathbf{v}}_{t-1}, \mathbf{v}_t) \qquad \text{(Amsgrad)}$$

The definitions of $\mathbf{m}_t$ and $\mathbf{v}_t$ are the same as Adam. Note that the modified $\widehat{\mathbf{v}}_t$ ensures the convergence of Amsgrad. Reddi et al. (2018) also corrected some mistakes in the original proof of Adam and proved an $O(1/\sqrt{T})$ convergence rate of Amsgrad for convex optimization. Note that all the theoretical guarantees on adaptive gradient methods (Adagrad, Adam, Amsgrad) are only proved for convex functions.

## 3 THE PROPOSED ALGORITHM

In this section, we propose a new algorithm that not only inherits the $O(1/\sqrt{T})$ convergence rate from Adam/Amsgrad, but also has comparable or even better generalization performance than SGD with momentum. Specifically, we introduce the a partial adaptive parameter $p$ to control the level of adaptivity of the optimization procedure. The proposed algorithm is displayed in in Algorithm 1.

---

**Algorithm 1** Partially adaptive momentum estimation method (Padam)

---

    **input:** initial point $\boldsymbol{\theta}_1 \in \mathcal{X}$; step sizes $\{\alpha_t\}$; momentum parameters $\{\beta_{1t}\}, \beta_2$; partially adaptive parameter $p \in (0, 1/2]$
    set $\mathbf{m}_0 = \mathbf{0}, \mathbf{v}_0 = \mathbf{0}, \widehat{\mathbf{v}}_0 = \mathbf{0}$
    **for** $t = 1, \ldots, T$ **do**
        $\mathbf{g}_t = \nabla f_t(\boldsymbol{\theta}_t)$
        $\mathbf{m}_t = \beta_{1t}\mathbf{m}_{t-1} + (1-\beta_{1t})\mathbf{g}_t$
        $\mathbf{v}_t = \beta_2\mathbf{v}_{t-1} + (1-\beta_2)\mathbf{g}_t^2$
        $\widehat{\mathbf{v}}_t = \max(\widehat{\mathbf{v}}_{t-1}, \mathbf{v}_t)$
        $\boldsymbol{\theta}_{t+1} = \Pi_{\mathcal{X}, \text{diag}(\widehat{\mathbf{v}}_t^p)}\big(\boldsymbol{\theta}_t - \alpha_t \cdot \mathbf{m}_t/\widehat{\mathbf{v}}_t^p\big)$
    **end for**

---

In Algorithm 1, $\mathbf{g}_t$ denotes the stochastic gradient and $\widehat{\mathbf{v}}_t$ can be seen as a moving average over the second order momentum of the stochastic gradients. As we can see from Algorithm 1, the key

---

    [2]Here for simplicity and consistency, we ignore the bias correction step in the original paper of Adam. Yet adding the bias correction step will not affect the argument in the paper.

difference between Padam and Amsgrad (Reddi et al., 2018) is that: while $\mathbf{m}_t$ is still the first order momentum as in Adam/Amsgrad, it is now "partially adapted" by the second order momentum, i.e.,

$$\boldsymbol{\theta}_{t+1} = \boldsymbol{\theta}_t - \alpha_t \frac{\mathbf{m}_t}{\widehat{\mathbf{v}}_t^p}, \text{ where } \widehat{\mathbf{v}}_t = \max(\widehat{\mathbf{v}}_{t-1}, \mathbf{v}_t) \qquad \text{(Padam)}$$

We call $p \in (0, 1/2]$ the partially adaptive parameter. Note that $1/2$ is the largest possible value for $p$ and a larger $p$ will result in non-convergence in the proof (see the details of the proof in the appendix). When $p \to 0$, Algorithm 1 reduces to SGD with momentum[3] and when $p = 1/2$, Algorithm 1 is exactly Amsgrad. Therefore, Padam indeed unifies Amsgrad and SGD with momentum.

Now the question is, what value for $p$ should we choose? Or in another way, is $p = 1/2$ the best choice? The answer is negative. The intuition behind this is simple: it is very likely that Adam/Amsgrad is "over-adaptive". One notable fact that people found when using adaptive gradient methods to train deep neural networks is that the learning rate needs to be much smaller than that of SGD-momentum (Keskar & Socher, 2017; Wilson et al., 2017). For many tasks, the base learning rate for SGD-momentum is usually set to be $0.1$ while that of Adam is usually set to be $0.001$. In fact, the key reason that prohibits Adam from adopting a more aggressive learning rate is the large adaptive term $1/\sqrt{\widehat{\mathbf{v}}_t}$. The existence of such a large adaptive term makes the effective learning rate ($\alpha_t/\sqrt{\widehat{\mathbf{v}}_t}$) easily explode with a larger $\alpha_t$. Moreover, the learning rate decaying strategy used in modern deep neural network training makes things worse. More specifically, after several rounds of decaying, the learning rates of the adaptive gradient methods are too small to make any significant progress in the training process. We call it "small learning rate dilemma". This explains the relatively weak performances of adaptive gradient methods at the later stage of the training process, where the non-adaptive gradient methods like SGD start to outperform them.

The above discussion suggests that we should consider using Padam with a proper adaptive parameter $p$, which will enable us to adopt a larger learning rate to avoid the "small learning rate dilemma". And we will show in our experiments (Section 5) that Padam can adopt an equally large base learning rate as SGD with momentum.

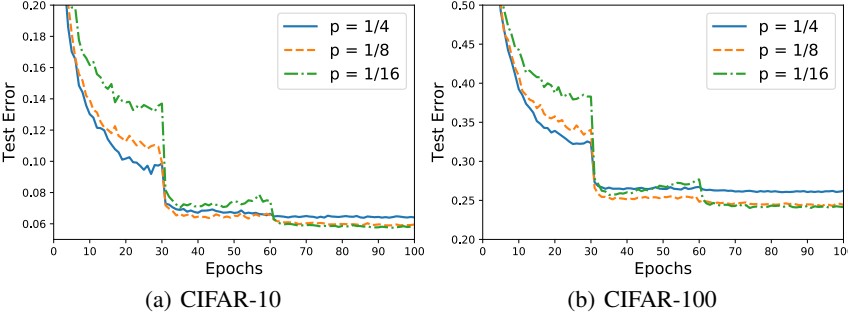

(a) CIFAR-10          (b) CIFAR-100

Figure 1: Performance comparison of Padam with different choices of $p$ for training ResNet on CIFAR-10 / CIFAR-100 dataset.

Figure 1 shows the comparison of test error performances under the different partial adaptive parameter $p$ for ResNet on both CIFAR-10 and CIFAR-100 datasets. We can observe that a larger $p$ will lead to fast convergence at early stages and worse generalization performance later, while a smaller $p$ behaves more like SGD with momentum: slow in early stages but finally catch up. With a proper choice of $p$ (e.g., $1/8$ in this case), Padam can obtain the best of both worlds.

Note that besides Algorithm 1, our partially adaptive idea can also be applied to other adaptive gradient methods such as Adagrad, Adadelta, RMSprop, AdaMax (Kingma & Ba, 2015). For the sake of conciseness, we do not list the partially adaptive versions for other adaptive gradient methods here. We also would like to comment that Padam is totally different from the $p$-norm generalized version of Adam in Kingma & Ba (2015), which induces AdaMax method when $p \to \infty$. In their

---

[3]The only difference between Padam with $p = 0$ and SGD with momentum is an extra constant factor $(1 - \beta_1)$, which can be moved into the learning rate such that the update rules for these two algorithms are identical.

case, $p$-norm is used to generalize 2-norm of their current and past gradients while keeping the scale of adaptation unchanged. In sharp contrast, we intentionally change (reduce) the scale of the adaptive term in Padam to get better generalization performance.

Finally, note that in Algorithm 1 we remove the bias correction step used in the original Adam paper following Reddi et al. (2018). Nevertheless, our arguments and theory are applicable to the bias correction version as well.

# 4 CONVERGENCE ANALYSIS OF THE PROPOSED ALGORITHM

In this section, we establish the theory of convergence for our proposed algorithm in the online optimization setting (Zinkevich, 2003), where we try to minimize the cumulative objective value of a sequence of loss functions: $f_1, f_2, \ldots, f_T$. In particular, at each time step $t$, the optimization algorithm picks a point $\boldsymbol{\theta}_t \in \mathcal{X}$, where $\mathcal{X} \in \mathbb{R}^d$ is the feasible set. A loss function $f_t$ is then revealed, and the algorithm incurs loss $f_t(\boldsymbol{\theta}_t)$. Let $\boldsymbol{\theta}^*$ be optimal solution to the cumulative objective function as follows

$$\boldsymbol{\theta}^* \in \operatorname*{argmin}_{\boldsymbol{\theta} \in \mathcal{X}} \sum_{t=1}^T f_t(\boldsymbol{\theta}),$$

where $\mathcal{X}$ is a feasible set for all steps. We evaluate our algorithm using the regret, which characterizes the sum of all previous loss function values $f_t(\boldsymbol{\theta}_t)$ relative to the performance of the best fixed parameter $\boldsymbol{\theta}^*$ from a feasible set. Specifically, the regret is defined as

$$R_T = \sum_{t=1}^T \big(f_t(\boldsymbol{\theta}_t) - f_t(\boldsymbol{\theta}^*)\big),$$

and our goal is to predict the unknown parameter $\boldsymbol{\theta}_t$ and minimize the overall regret $R_T$. Our theory is established for convex loss functions, where the following assumption holds.

**Assumption 4.1** (Convex function). All $f_t(\boldsymbol{\theta})$ are convex functions on $\mathcal{X}$ for $1 \leq t \leq T$, i.e., for all $\mathbf{x}, \mathbf{y} \in \mathcal{X}$,

$$f_t(\mathbf{y}) \geq f_t(\mathbf{x}) + \nabla f_t(\mathbf{x})^\top (\mathbf{y} - \mathbf{x}).$$

Assumption 4.1 is a standard assumption in online learning and the same assumption has been used in the analysis of Adagrad (Duchi et al., 2011), Adam (Kingma & Ba, 2015) and Amsgrad (Reddi et al., 2018).

Next we provide the main convergence rate result for our proposed algorithm.

**Theorem 4.2.** Under Assumption 4.1, if the convex feasible set $\mathcal{X}$ has bounded diameters, i.e., $\|\boldsymbol{\theta} - \boldsymbol{\theta}^*\|_\infty \leq D_\infty$ for all $\boldsymbol{\theta} \in \mathcal{X}$, and $f_t$ has bounded gradients, i.e., $\|\nabla f_t(\boldsymbol{\theta})\|_\infty \leq G_\infty$ for all $\boldsymbol{\theta} \in \mathcal{X}, 1 \leq t \leq T$, let $\alpha_t = \alpha/\sqrt{t}, \beta_{1t} = \beta_1 \lambda^{t-1}, \lambda \in (0,1), 0 \leq \beta_1, \beta_2 < 1, p \in (0, 0.5]$, the regret of Algorithm 1 satisfies:

$$R_T \leq \frac{D_\infty^2}{2\alpha(1-\beta_1)} \sum_{i=1}^d \sqrt{T} \cdot \widehat{v}_{T,i}^p + \frac{\alpha G_\infty^{(1-2p)} \sqrt{1 + \log T}}{(1-\beta_1)^2 (1-\gamma)(1-\beta_2)^p} \sum_{i=1}^d \|g_{1:T,i}\|_2$$
$$+ \frac{\beta_1 d D_\infty^2 G_\infty^{2p}}{2\alpha(1-\beta_1)(1-\lambda)^2}, \tag{4.1}$$

where $\gamma = \beta_1/\sqrt{\beta_2} < 1$.

**Remark 4.3.** Theorem 4.2 suggests that, similar to Adam (Kingma & Ba, 2015) and Amsgrad (Reddi et al., 2018), the regret of Padam can be considerably better than online gradient descent (which is known to have a regret bound of $O(\sqrt{dT})$) when $\sum_{i=1}^d \|g_{1:T,i}\|_2 \ll \sqrt{dT}$ (Duchi et al., 2011) and $\sum_{i=1}^d \widehat{v}_{T,i}^p \ll \sqrt{d}$. Also, from the proof of Theorem 4.2 in Appendix A, we can see that the regret bound can remain in the same order even with a more modest momentum decay $\beta_{1t} = \beta_1/t$.

The following corollary demonstrates that our proposed algorithm enjoys a regret bound of $\widetilde{O}(\sqrt{T})$, which is comparable to the best known bound for general convex online learning problems.

**Corollary 4.4.** Under the same conditions of Theorem 4.2, for all $T \geq 1$, the regret of Algorithm 1 satisfies $R_T = \widetilde{O}(\sqrt{T})$.

Corollary 4.4 suggests that Padam attains $R_T = o(T)$ for all situations (no matter whether the data features are sparse or not). This suggests that Algorithm 1 indeed converges to the optimal solution when the loss functions are convex, as shown by the fact that $\lim_{T \to \infty} R_T / T \to 0$.

## 5 EXPERIMENTS

In this section, we empirically evaluate our proposed algorithm for training various modern deep learning models and test them on several standard benchmarks[4]. We show that for nonconvex loss functions in deep learning, our proposed algorithm still enjoys a fast convergence rate, while its generalization performance is as good as SGD with momentum and much better than existing adaptive gradient methods such as Adam and Amsgrad.

### 5.1 ENVIRONMENTAL SETUP

All experiments are conducted on Amazon AWS p3.8xlarge servers which come with Intel Xeon E5 CPU and 4 NVIDIA Tesla V100 GPUs (16G RAM per GPU). All experiments are implemented in Pytorch platform version $0.4.1$ within Python $3.6.4$.

### 5.2 BASELINE ALGORITHMS

We compare our proposed algorithm against several state-of-the-art algorithms, including: (1) SGD with momentum, (2) Adam (Kingma & Ba, 2015), (3) Amsgrad (Reddi et al., 2018), and (4) AdamW (Loshchilov & Hutter, 2017).

Note that we do not perform the projection step explicitly in all experiments as in Reddi et al. (2018). Also both Amsgrad and Padam will perform the bias correction step as in Adam for a fair comparison.

### 5.3 DATASETS / PARAMETER SETTINGS / CNN ARCHITECTURES

We use several popular datasets for image classifications: CIFAR-10 (Krizhevsky & Hinton, 2009), CIFAR-100 (Krizhevsky & Hinton, 2009), ImageNet dataset (ILSVRC2012) (Deng et al., 2009). We adopt three popular CNN architectures to evaluate the performance of our proposed algorithms: VGGNet (Simonyan & Zisserman, 2014), Residual Neural Network (ResNet) (He et al., 2016), Wide Residual Network(Zagoruyko & Komodakis, 2016). We test CIFAR-10 and CIFAR-100 tasks for 200 epochs. For all experiments, we use a fixed multi-stage learning rate decaying scheme: we decay the learning rate by $0.1$ at the 50th, 100th and 150th epochs. We test ImageNet tasks for 100 epochs with the same multi-stage learning rate decaying scheme at the 30th, 60th and 80th epochs. We perform grid search on validation set to choose the best base learning rate for all algorithms and also the second order momentum parameter $\beta_2$ for all the adaptive gradient methods. In terms of the choice of $p$, we recommend doing binary search from $1/4$ to $1/8$, to $1/16$, etc. Yet in most cases we tested, $1/8$ is a stable and reliable choice of $p$.

Details about the datasets, CNN architectures and all the specific model parameters used in the following experiments can be found in the supplementary materials.

### 5.4 EXPERIMENTAL RESULTS

We compare our proposed algorithm with other baselines on training the aforementioned three modern CNN architectures for image classification on the CIFAR-10, CIFAR-100 and also ImageNet

---

[4]The code to reproduce the experimental results is available online. The URL is removed for double-blind review

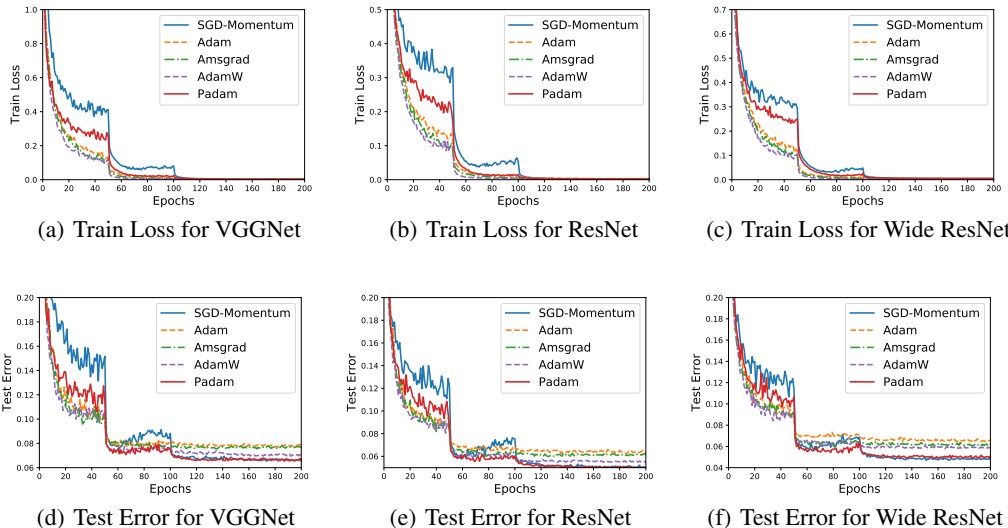

|                    |                    |                    |
|--------------------|--------------------|--------------------|
| (a) Train Loss for VGGNet | (b) Train Loss for ResNet | (c) Train Loss for Wide ResNet |
| (d) Test Error for VGGNet | (e) Test Error for ResNet | (f) Test Error for Wide ResNet |

Figure 2: Train loss and test error (top-1 error) of three CNN architectures on Cifar-10. In all cases, Padam achieves the fastest training procedure among all methods and generalizes slightly better than SGD-momentum.

datasets. Due to the paper length limit, we leave all our experimental results regarding CIFAR-100 dataset in the supplementary materials. Figure 2 plots the train loss and test error (top-1 error) against training epochs on the CIFAR-10 dataset. As we can see from the figure, at the early stage of the training process, (partially) adaptive gradient methods including Padam, make rapid progress lowing both the train loss and the test error, while SGD with momentum converges relatively slowly. After the first learning rate decaying at the 50-th epoch, different algorithms start to behave differently. SGD with momentum makes a huge drop while fully adaptive gradient methods (Adam and Amsgrad, AdamW) start to generalize badly. Padam, on the other hand, maintains relatively good generalization performance and also holds the lead over other algorithms. After the second decaying at the 100-th epoch, Adam and Amsgrad basically lose all the power of traversing through the parameter space due to the "small learning dilemma", while the performance of SGD with momentum finally catches up with Padam. AdamW improves the performance compared with original Adam but there are still generalization gaps left behind, at least in our test settings. Overall we can see that Padam indeed achieves the best of both worlds (i.e., Adam and SGD with momentum): it maintains faster convergence rate while also generalizing as well as SGD with momentum in the end. Table 1 shows the test accuracy of VGGNet on CIFAR-10 dataset, from which we can also observe the fast convergence of Padam. Specifically, Padam achieves the best test accuracy at 100th, 150th and 200th epochs for CIFAR-10, except for the 50th, where fully adaptive methods like Adam, Amsgrad and AdamW still enjoy their fast early stage convergence performances. This suggests that practitioners should once again, use fast to converge partially adaptive gradient methods for training deep neural networks, without worrying about the generalization performances.

Table 1: Test accuracy of VGGNet on CIFAR-10. Bold number indicates the best result.

| Methods | Test Accuracy (%) | | | |
|---------|-----------|------------|------------|------------|
|         | 50th Epoch | 100th Epoch | 150th Epoch | 200th Epoch |
| SGD Momentum | $84.83 \pm 0.43$ | $91.18 \pm 0.14$ | $93.27 \pm 0.10$ | $93.32 \pm 0.08$ |
| Adam | $89.37 \pm 0.37$ | $91.83 \pm 0.23$ | $92.17 \pm 0.20$ | $92.21 \pm 0.06$ |
| Amsgrad | $\mathbf{90.25 \pm 0.39}$ | $92.25 \pm 0.18$ | $92.36 \pm 0.11$ | $92.36 \pm 0.09$ |
| AdamW | $89.36 \pm 0.29$ | $90.23 \pm 0.22$ | $92.86 \pm 0.15$ | $92.97 \pm 0.08$ |
| Padam | $88.47 \pm 0.55$ | $\mathbf{92.49 \pm 0.16}$ | $\mathbf{93.28 \pm 0.12}$ | $\mathbf{93.45 \pm 0.12}$ |

Figure 3 plots the Top-1 and Top-5 error against training epochs on the ImageNet dataset for both VGGNet and ResNet[5]. We can see that on ImageNet, all methods behave similarly as in our CIFAR-10 experiments. Padam method again obtains the best from both worlds by achieving the fastest convergence while generalizing as well as SGD with momentum. Even though Amsgrad and AdamW have greatly improve the performance comparing with standard Adam, it still suffers from the generalization gap on ImageNet.

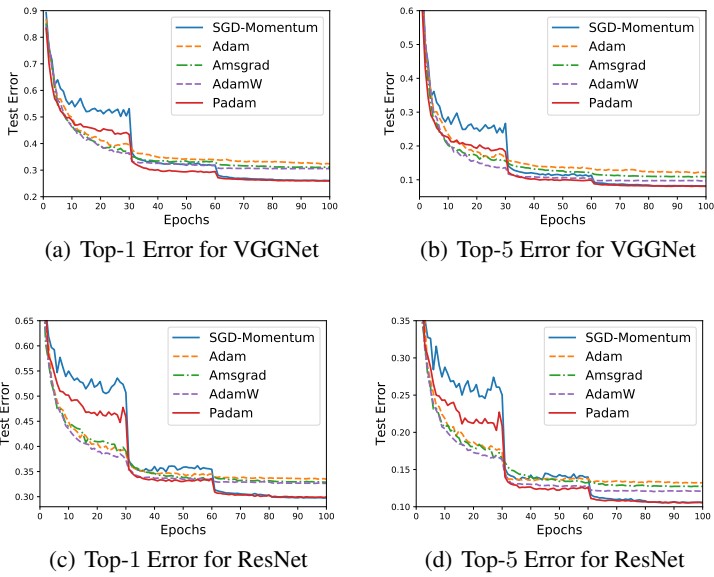

(a) Top-1 Error for VGGNet

(b) Top-5 Error for VGGNet

(c) Top-1 Error for ResNet

(d) Top-5 Error for ResNet

Figure 3: Top-1 and Top-5 error for VGGNet and ResNet on ImageNet dataset. In all cases, Padam achieves the fastest training speed and generalizes as well as SGD with momentum.

## 6 RELATED WORK

We briefly review the related work in this section. There are only several studies closely related to improving the generalization performance of Adam. Zhang et al. (2017) proposed a normalized direction-preserving Adam (ND-Adam), which changes the adaptive terms from individual dimensions to the whole gradient vector. This makes it more like a SGD-momentum with learning rate adjusted in each iteration, rather than an adaptive gradient algorithm. The empirical result also shows that its performance resembles SGD with momentum. Keskar & Socher (2017) proposed to improve the generalization performance by switching from Adam to SGD. Yet the empirical result shows that it actually sacrifices some of the convergence speed for better generalization rather than achieving the best from both worlds. Also deciding the switching learning rate and the best switching point requires extra efforts on parameter tuning since they can be drastically different for different tasks according to the paper. Loshchilov & Hutter (2017) proposed to fix the weight decay regularization in Adam by decoupling the weight decay from the gradient update and this improves the generalization performance of Adam.

## 7 CONCLUSIONS AND FUTURE WORK

In this paper, we proposed Padam, which unifies Adam/Amsgrad with SGD-momentum. With an appropriate choice of the partially adaptive parameter, we show that Padam can achieve the best from both worlds, i.e., maintaining fast convergence rate while closing the generalization gap. We also provide a theoretical analysis towards the convergence rate of Padam and show a similar data-dependent regret bound as in Duchi et al. (2011); Reddi et al. (2018).

---

[5]We did not conduct WideResNet experiment on Imagenet dataset due to GPU memory limits.

It would be interesting to see how well Padam performs in other types of neural networks, such as recurrent neural networks (RNNs) (Hochreiter & Schmidhuber, 1997) and generative adversarial networks (GANs) (Goodfellow et al., 2014). We leave it as a future work.

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

# A PROOF OF THE MAIN RESULTS

## A.1 PROOF OF THEOREM 4.2

The proof of Theorem 4.2 needs the following lemmas.

**Lemma A.1.** Under the same conditions as in Theorem 4.2, we have

$$\sum_{t=1}^{T}\sum_{i=1}^{d}\frac{\alpha_t \cdot m_{t,i}^2}{\widehat{v}_{t,i}^p} \leq \frac{\alpha G_\infty^{(1-2p)}\sqrt{1+\log T}}{(1-\beta_1)(1-\gamma)(1-\beta_2)^p}\sum_{i=1}^{d}\|g_{1:T,i}\|_2. \tag{A.1}$$

Lemma A.1 basically describes the bound of the key quantity $(\sum_{t=1}^{T}\sum_{i=1}^{d}\alpha_t \cdot m_{t,i}^2/\widehat{v}_{t,i}^p)$ in the convergence analysis of Algorithm 1. Its proof is inspired by Lemma 2 in Reddi et al. (2018). Note that the original proof in Kingma & Ba (2015) for the corresponding quantity contains some mistakes, which result in a bound without the logarithmic term in (A.1). Here we follow the corrected version in Reddi et al. (2018).

**Lemma A.2.** For any $\mathbf{V} \in \mathcal{S}_{++}^d$ and convex feasible set $\mathcal{X} \subset \mathbb{R}^d$, suppose $\mathbf{a}_1 = \Pi_{\mathcal{X},\mathbf{V}^p}(\mathbf{b}_1)$, $\mathbf{a}_2 = \Pi_{\mathcal{X},\mathbf{V}^p}(\mathbf{b}_2)$ for $p \in (0, 1/2]$, we have

$$\|\mathbf{V}^p(\mathbf{a}_1 - \mathbf{a}_2)\|_2 \leq \|\mathbf{V}^p(\mathbf{b}_1 - \mathbf{b}_2)\|_2, \tag{A.2}$$

where $p \in (0, 1/2]$ is an absolute constant.

Lemma A.2 is an adaptation of Lemma 5 in McMahan & Streeter (2010) (or Lemma 4 in Reddi et al. (2018)).

Now we are ready to prove Theorem 4.2.

*Proof of Theorem 4.2.* By Assumption 4.1, all $f_t$'s are convex, we have

$$\sum_{t=1}^{T} f_t(\boldsymbol{\theta}_t) - f_t(\boldsymbol{\theta}_t^*) \leq \sum_{t=1}^{T}\langle\mathbf{g}_t, \boldsymbol{\theta}_t - \boldsymbol{\theta}^*\rangle. \tag{A.3}$$

Consider the update rule for Algorithm 1, let $\widehat{\mathbf{V}}_t = \mathrm{diag}(\widehat{\mathbf{v}}_t)$, then we have $\boldsymbol{\theta}_{t+1} = \Pi_{\mathcal{X},\widehat{\mathbf{V}}_t^p}(\boldsymbol{\theta}_t - \alpha_t\widehat{\mathbf{V}}_t^{-p}\cdot\mathbf{m}_t)$. Since $\Pi_{\mathcal{X},\widehat{\mathbf{V}}_t^p}(\boldsymbol{\theta}^*) = \boldsymbol{\theta}^*$ for all $\boldsymbol{\theta}^* \in \mathcal{X}$, by Lemma A.2 we have

$$\left\|\widehat{\mathbf{V}}_t^{p/2}(\boldsymbol{\theta}_{t+1} - \boldsymbol{\theta}^*)\right\|_2^2 \leq \left\|\widehat{\mathbf{V}}_t^{p/2}(\boldsymbol{\theta}_t - \alpha_t\widehat{\mathbf{V}}_t^{-p}\cdot\mathbf{m}_t - \boldsymbol{\theta}^*)\right\|_2^2. \tag{A.4}$$

Now expand the square term on the right hand side of (A.4), we have

$$\left\|\widehat{\mathbf{V}}_t^{p/2}(\boldsymbol{\theta}_{t+1} - \boldsymbol{\theta}^*)\right\|_2^2 \leq \left\|\widehat{\mathbf{V}}_t^{p/2}(\boldsymbol{\theta}_t - \boldsymbol{\theta}^*)\right\|_2^2 + \alpha_t^2\left\|\widehat{\mathbf{V}}_t^{-p/2}\mathbf{m}_t\right\|_2^2 - 2\alpha_t\langle\mathbf{m}_t, \boldsymbol{\theta}_t - \boldsymbol{\theta}^*\rangle$$
$$= \left\|\widehat{\mathbf{V}}_t^{p/2}(\boldsymbol{\theta}_t - \boldsymbol{\theta}^*)\right\|_2^2 + \alpha_t^2\left\|\widehat{\mathbf{V}}_t^{-p/2}\mathbf{m}_t\right\|_2^2 - 2\alpha_t\langle\beta_{1t}\mathbf{m}_{t-1} + (1-\beta_{1t})\mathbf{g}_t, \boldsymbol{\theta}_t - \boldsymbol{\theta}^*\rangle, \tag{A.5}$$

where the second equality follows from the definition for $\mathbf{m}_t$. Rearrange the items in (A.5), we have

$$\langle\mathbf{g}_t, \boldsymbol{\theta}_t - \boldsymbol{\theta}^*\rangle = \frac{1}{2\alpha_t(1-\beta_{1t})}\left[\left\|\widehat{\mathbf{V}}_t^{p/2}(\boldsymbol{\theta}_{t+1} - \boldsymbol{\theta}^*)\right\|_2^2 - \left\|\widehat{\mathbf{V}}_t^{p/2}(\boldsymbol{\theta}_t - \boldsymbol{\theta}^*)\right\|_2^2\right] + \frac{\alpha_t}{2(1-\beta_{1t})}\cdot\left\|\widehat{\mathbf{V}}_t^{-p/2}\mathbf{m}_t\right\|_2^2$$
$$-\frac{\beta_{1t}}{1-\beta_{1t}}\langle\mathbf{m}_{t-1}, \boldsymbol{\theta}_t - \boldsymbol{\theta}^*\rangle$$
$$\leq \frac{1}{2\alpha_t(1-\beta_{1t})}\left[\left\|\widehat{\mathbf{V}}_t^{p/2}(\boldsymbol{\theta}_{t+1} - \boldsymbol{\theta}^*)\right\|_2^2 - \left\|\widehat{\mathbf{V}}_t^{p/2}(\boldsymbol{\theta}_t - \boldsymbol{\theta}^*)\right\|_2^2\right] + \frac{\alpha_t}{2(1-\beta_{1t})}\cdot\left\|\widehat{\mathbf{V}}_t^{-p/2}\mathbf{m}_t\right\|_2^2$$
$$+\frac{\beta_{1t}\alpha_{t-1}}{2(1-\beta_{1t})}\cdot\left\|\widehat{\mathbf{V}}_{t-1}^{-p/2}\mathbf{m}_{t-1}\right\|_2^2 + \frac{\beta_{1t}}{2\alpha_{t-1}(1-\beta_{1t})}\left\|\widehat{\mathbf{V}}_{t-1}^{p/2}(\boldsymbol{\theta}_t - \boldsymbol{\theta}^*)\right\|_2^2, \tag{A.6}$$

where the inequality holds due to Young's inequality. Summing over (A.6) for all $1 \le t \le T$ and submitting it back into (A.3), we obtain

$$
\sum_{t=1}^{T} f_t(\boldsymbol{\theta}_t) - f_t(\boldsymbol{\theta}_t^*)
$$

$$
\le \sum_{t=1}^{T} \frac{1}{2\alpha_t(1-\beta_{1t})} \left[ \left\| \widehat{\mathbf{V}}_t^{p/2}(\boldsymbol{\theta}_{t+1} - \boldsymbol{\theta}^*) \right\|_2^2 - \left\| \widehat{\mathbf{V}}_t^{p/2}(\boldsymbol{\theta}_t - \boldsymbol{\theta}^*) \right\|_2^2 \right] + \sum_{t=2}^{T} \frac{\beta_{1t}}{2\alpha_{t-1}(1-\beta_{1t})} \left\| \widehat{\mathbf{V}}_{t-1}^{p/2}(\boldsymbol{\theta}_t - \boldsymbol{\theta}^*) \right\|_2^2
$$

$$
+ \sum_{t=2}^{T} \frac{\beta_{1t}\alpha_{t-1}}{2(1-\beta_{1t})} \cdot \left\| \widehat{\mathbf{V}}_{t-1}^{-p/2}\mathbf{m}_{t-1} \right\|_2^2 + \sum_{t=1}^{T} \frac{\alpha_t}{2(1-\beta_{1t})} \cdot \left\| \widehat{\mathbf{V}}_t^{-p/2}\mathbf{m}_t \right\|_2^2
$$

$$
\le \underbrace{\sum_{t=1}^{T} \sum_{i=1}^{d} \frac{\widehat{v}_{t,i}^p}{2\alpha_t(1-\beta_1)} \left[ (\theta_{t+1,i} - \theta_i^*)^2 - (\theta_{t,i} - \theta_i^*)^2 \right]}_{I_1} + \underbrace{\sum_{t=1}^{T} \sum_{i=1}^{d} \frac{\beta_{1t} \cdot \widehat{v}_{t,i}^p}{2\alpha_t(1-\beta_1)} (\theta_{t+1,i} - \theta_i^*)^2}_{I_2}
$$

$$
+ \underbrace{\frac{1+\beta_1}{2(1-\beta_1)} \sum_{t=1}^{T} \sum_{i=1}^{d} \frac{\alpha_t \cdot m_{t,i}^2}{\widehat{v}_{t,i}^p}}_{I_3}, \tag{A.7}
$$

where the last inequality holds due to the fact that $\beta_{1t} = \beta_1 \lambda^{t-1}$ is monotonically decreasing with $t$. For term $I_1$, we have

$$
\sum_{t=1}^{T} \sum_{i=1}^{d} \frac{\widehat{v}_{t,i}^p}{2\alpha_t(1-\beta_1)} \left[ (\theta_{t+1,i} - \theta_i^*)^2 - (\theta_{t,i} - \theta_i^*)^2 \right]
$$

$$
= \frac{1}{2(1-\beta_1)} \sum_{i=1}^{d} \frac{\widehat{v}_{1,i}^p \cdot (\theta_{1,i} - \theta_i^*)^2}{\alpha_1} + \frac{1}{2(1-\beta_1)} \sum_{t=2}^{T} \sum_{i=1}^{d} \left( \frac{\widehat{v}_{t,i}^p}{\alpha_t} - \frac{\widehat{v}_{t-1,i}^p}{\alpha_{t-1}} \right) (\theta_{t,i} - \theta_i^*)^2
$$

$$
\le \frac{D_\infty^2}{2(1-\beta_1)} \left( \sum_{i=1}^{d} \frac{\widehat{v}_{1,i}^p}{\alpha_1} + \sum_{t=2}^{T} \sum_{i=1}^{d} \left( \frac{\widehat{v}_{t,i}^p}{\alpha_t} - \frac{\widehat{v}_{t-1,i}^p}{\alpha_{t-1}} \right) \right)
$$

$$
= \frac{D_\infty^2}{2(1-\beta_1)} \sum_{i=1}^{d} \frac{\widehat{v}_{T,i}^p}{\alpha_T}
$$

$$
= \frac{D_\infty^2}{2\alpha(1-\beta_1)} \sum_{i=1}^{d} \sqrt{T} \cdot \widehat{v}_{T,i}^p, \tag{A.8}
$$

where the inequality follows from the bounded diameter condition and the definition of $\widehat{\mathbf{v}}_t$ ensures that $\widehat{v}_{t,i}^p/\alpha_t - \widehat{v}_{t-1,i}^p/\alpha_{t-1} > 0$, and the second equality holds due to simple telescope sum. For term $I_2$, note that simple calculation yields that $\widehat{v}_{t,i}^p \le G_\infty^{2p}$, we have

$$
\sum_{t=1}^{T} \sum_{i=1}^{d} \frac{\beta_{1t}\widehat{v}_{t,i}^p}{2\alpha_t(1-\beta_1)} (\theta_{t+1,i} - \theta_i^*)^2 \le \frac{\beta_1 d D_\infty^2 G_\infty^{2p}}{2\alpha(1-\beta_1)} \sum_{t=1}^{T} \sqrt{t}\lambda^{t-1} \le \frac{\beta_1 d D_\infty^2 G_\infty^{2p}}{2\alpha(1-\beta_1)(1-\lambda)^2}, \tag{A.9}
$$

where the last inequality follows from first relaxing $\sqrt{t}$ to $t$ and then using the geometric series summation rule. For term $I_3$, by Lemma A.1 we have

$$
\frac{1+\beta_1}{2(1-\beta_1)} \sum_{t=1}^{T} \sum_{i=1}^{d} \frac{\alpha_t \cdot m_{t,i}^2}{\widehat{v}_{t,i}^p} \le \frac{\alpha G_\infty^{(1-2p)}\sqrt{1+\log T}}{(1-\beta_1)^2(1-\gamma)(1-\beta_2)^p} \sum_{i=1}^{d} \|g_{1:T,i}\|_2. \tag{A.10}
$$

Submitting (A.8), (A.9) and (A.10) back into (A.7) yields the desired result. □

## A.2 PROOF OF COROLLARY 4.4

*Proof of Corollary 4.4.* By Theorem 4.2, we have

$$R_T \leq \frac{D_\infty^2}{2\alpha(1-\beta_1)} \sum_{i=1}^{d} \sqrt{T} \cdot \widehat{v}_{T,i}^p + \frac{\alpha G_\infty^{(1-2p)} \sqrt{1+\log T}}{(1-\beta_1)^2(1-\gamma)(1-\beta_2)^p} \sum_{i=1}^{d} \|g_{1:T,i}\|_2$$
$$+ \frac{\beta_1 d D_\infty^2 G_\infty^{2p}}{2\alpha(1-\beta_1)(1-\lambda)^2}.$$

Note that we have

$$\sum_{i=1}^{d} \|g_{1:T,i}\|_2 \leq \sum_{i=1}^{d} \sqrt{\sum_{t=1}^{T} g_{t,i}^2} \leq d G_\infty \sqrt{T},$$

and by definition we also have

$$\widehat{v}_{t,i}^p = \left( (1-\beta_2) \sum_{j=1}^{T} \beta_2^{T-j} g_{j,i}^2 \right)^p \leq (1-\beta_2)^p \cdot G_\infty^{2p} \left( \sum_{j=1}^{T} \beta_2^{T-j} \right)^p \leq G_\infty^{2p}.$$

Combine the above results we can easily have

$$R_T = \widetilde{O}\big(\sqrt{T}\big).$$

This completes the proof. ☐

# B PROOF OF TECHNICAL LEMMAS IN APPENDIX A

## B.1 PROOF OF LEMMA A.1

*Proof.* Consider

$$\sum_{t=1}^{T} \sum_{i=1}^{d} \frac{\alpha_t \cdot m_{t,i}^2}{\widehat{v}_{t,i}^p} = \sum_{t=1}^{T-1} \sum_{i=1}^{d} \frac{\alpha_t \cdot m_{t,i}^2}{\widehat{v}_{t,i}^p} + \sum_{i=1}^{d} \frac{\alpha_T \cdot m_{T,i}^2}{\widehat{v}_{T,i}^p}$$
$$\leq \sum_{t=1}^{T-1} \sum_{i=1}^{d} \frac{\alpha_t \cdot m_{t,i}^2}{\widehat{v}_{t,i}^p} + \sum_{i=1}^{d} \frac{\alpha_T \cdot m_{T,i}^2}{v_{T,i}^p}$$
$$\leq \sum_{t=1}^{T-1} \sum_{i=1}^{d} \frac{\alpha_t \cdot m_{t,i}^2}{\widehat{v}_{t,i}^p} + \frac{\alpha}{\sqrt{T}} \sum_{i=1}^{d} \frac{\left( \sum_{j=1}^{T} (1-\beta_{1j})\beta_1^{T-j} g_{j,i} \right)^2}{\left( (1-\beta_2) \sum_{j=1}^{T} \beta_2^{T-j} g_{j,i}^2 \right)^p}, \qquad \text{(B.1)}$$

where the first inequality holds due to the definition of $\widehat{v}_{T,i}$ and the second inequality follows from the fact that $\beta_{1t} = \beta_1 \lambda^{t-1}$ and $\lambda \in (0,1)$. (B.1) can be further bounded as:

$$\sum_{t=1}^{T} \sum_{i=1}^{d} \frac{\alpha_t \cdot m_{t,i}^2}{\widehat{v}_{t,i}^p} \leq \sum_{t=1}^{T-1} \sum_{i=1}^{d} \frac{\alpha_t \cdot m_{t,i}^2}{\widehat{v}_{t,i}^p} + \frac{\alpha}{\sqrt{T}(1-\beta_2)^p} \sum_{i=1}^{d} \frac{\left( \sum_{j=1}^{T} \beta_1^{T-j} g_{j,i}^{(1-2p)} \right)\left( \sum_{j=1}^{T} \beta_1^{T-j} g_{j,i}^{(1+2p)} \right)}{\left( \sum_{j=1}^{T} \beta_2^{T-j} g_{j,i}^2 \right)^p}$$
$$\leq \sum_{t=1}^{T-1} \sum_{i=1}^{d} \frac{\alpha_t \cdot m_{t,i}^2}{\widehat{v}_{t,i}^p} + \frac{\alpha G_\infty^{(1-2p)}}{\sqrt{T}(1-\beta_1)(1-\beta_2)^p} \sum_{i=1}^{d} \frac{\left( \sum_{j=1}^{T} \beta_1^{T-j} g_{j,i}^{(1+2p)} \right)}{\left( \sum_{j=1}^{T} \beta_2^{T-j} g_{j,i}^2 \right)^p}$$
$$\leq \sum_{t=1}^{T-1} \sum_{i=1}^{d} \frac{\alpha_t \cdot m_{t,i}^2}{\widehat{v}_{t,i}^p} + \frac{\alpha G_\infty^{(1-2p)}}{\sqrt{T}(1-\beta_1)(1-\beta_2)^p} \sum_{j=1}^{T} \sum_{i=1}^{d} \frac{\left( \beta_1^{T-j} g_{j,i}^{(1+2p)} \right)}{\left( \beta_2^{T-j} g_{j,i}^2 \right)^p}$$
$$\leq \sum_{t=1}^{T-1} \sum_{i=1}^{d} \frac{\alpha_t \cdot m_{t,i}^2}{\widehat{v}_{t,i}^p} + \frac{\alpha G_\infty^{(1-2p)}}{\sqrt{T}(1-\beta_1)(1-\beta_2)^p} \sum_{j=1}^{T} \sum_{i=1}^{d} \gamma^{T-j} |g_{j,i}|,$$

where the first inequality holds due to Cauchy-Schwarz inequality and the fact that $0 \leq \beta_{1j} < 1$, the second inequality follows from the bounded gradient condition and $\sum_{j=1}^{T} \beta_1^{T-j} \leq 1/(1-\beta_1)$,

and the last inequality is due to the fact that $\beta_1/\beta_2^p \leq \beta_1/\sqrt{\beta_2} = \gamma$. Now repeat the above process for $1 \leq t \leq T$, we have

$$
\sum_{t=1}^{T}\sum_{i=1}^{d}\frac{\alpha_t \cdot m_{t,i}^2}{\widehat{v}_{t,i}^p} \leq \frac{\alpha G_\infty^{(1-2p)}}{(1-\beta_1)(1-\beta_2)^p}\sum_{t=1}^{T}\frac{1}{\sqrt{t}}\sum_{j=1}^{t}\sum_{i=1}^{d}\gamma^{t-j}|g_{j,i}|
$$

$$
= \frac{\alpha G_\infty^{(1-2p)}}{(1-\beta_1)(1-\beta_2)^p}\sum_{i=1}^{d}\sum_{t=1}^{T}|g_{t,i}|\sum_{j=t}^{T}\frac{\gamma^{j-t}}{\sqrt{j}}
$$

$$
\leq \frac{\alpha G_\infty^{(1-2p)}}{(1-\beta_1)(1-\beta_2)^p}\sum_{i=1}^{d}\sum_{t=1}^{T}\frac{|g_{t,i}|}{\sqrt{t}}\sum_{j=t}^{T}\gamma^{j-t}
$$

$$
\leq \frac{\alpha G_\infty^{(1-2p)}}{(1-\beta_1)(1-\beta_2)^p}\sum_{i=1}^{d}\sum_{t=1}^{T}\frac{|g_{t,i}|}{(1-\gamma)\sqrt{t}}, \tag{B.2}
$$

where the equality holds due to the change of the order of summation, the last inequality follows from the fact that $\sum_{j=t}^{T}\gamma^{j-t} \leq 1/(1-\gamma)$. By Cauchy-Schwarz inequality , (B.2) can be further written as

$$
\sum_{t=1}^{T}\sum_{i=1}^{d}\frac{\alpha_t \cdot m_{t,i}^2}{\widehat{v}_{t,i}^p} \leq \frac{\alpha G_\infty^{(1-2p)}}{(1-\beta_1)(1-\gamma)(1-\beta_2)^p}\sum_{i=1}^{d}\|g_{1:T,i}\|_2\sqrt{\sum_{t=1}^{T}\frac{1}{t}}
$$

$$
\leq \frac{\alpha G_\infty^{(1-2p)}\sqrt{1+\log T}}{(1-\beta_1)(1-\gamma)(1-\beta_2)^p}\sum_{i=1}^{d}\|g_{1:T,i}\|_2.
$$

This completes the proof. $\qquad\square$

### B.2 PROOF OF LEMMA A.2

*Proof.* The proof is inspired from Lemma 4 in Reddi et al. (2018) and Lemma 5 in McMahan & Streeter (2010). Since $\mathbf{a}_1 = \mathrm{argmin}_{\mathbf{x}\in\mathcal{X}}\|\mathbf{V}^p(\mathbf{x}-\mathbf{b}_1)\|_2$ and $\mathbf{a}_2 = \mathrm{argmin}_{\mathbf{x}\in\mathcal{X}}\|\mathbf{V}^p(\mathbf{x}-\mathbf{b}_2)\|_2$, by projection property we have

$$
\langle\mathbf{V}^{2p}(\mathbf{a}_1-\mathbf{b}_1),\mathbf{a}_2-\mathbf{a}_1\rangle \geq 0,
$$

$$
\langle\mathbf{V}^{2p}(\mathbf{a}_2-\mathbf{b}_2),\mathbf{a}_1-\mathbf{a}_2\rangle \geq 0.
$$

Combine the above inequalities we have

$$
\langle\mathbf{V}^{2p}(\mathbf{a}_1-\mathbf{b}_1),\mathbf{a}_2-\mathbf{a}_1\rangle - \langle\mathbf{V}^{2p}(\mathbf{a}_2-\mathbf{b}_2),\mathbf{a}_2-\mathbf{a}_1\rangle \geq 0. \tag{B.3}
$$

Rearrange (B.3) yields that

$$
(\mathbf{b}_2-\mathbf{b}_1)^\top\mathbf{V}^{2p}(\mathbf{a}_2-\mathbf{a}_1) \geq (\mathbf{a}_2-\mathbf{a}_1)^\top\mathbf{V}^{2p}(\mathbf{a}_2-\mathbf{a}_1). \tag{B.4}
$$

Also since $\mathbf{V}$ is positive definite, the fact that $(\mathbf{b}_2-\mathbf{b}_1-(\mathbf{a}_2-\mathbf{a}_1))^\top\mathbf{V}^{2p}(\mathbf{b}_2-\mathbf{b}_1-(\mathbf{a}_2-\mathbf{a}_1)) \geq 0$ immediately implies that

$$
(\mathbf{b}_2-\mathbf{b}_1)^\top\mathbf{V}^{2p}(\mathbf{b}_2-\mathbf{b}_1) \geq -(\mathbf{a}_2-\mathbf{a}_1)^\top\mathbf{V}^{2p}(\mathbf{a}_2-\mathbf{a}_1) + 2(\mathbf{a}_2-\mathbf{a}_1)^\top\mathbf{V}^{2p}(\mathbf{b}_2-\mathbf{b}_1). \tag{B.5}
$$

Combining (B.4) and (B.5) we have the desired result. $\qquad\square$

## C EXPERIMENT DETAILS

### C.1 DATASETS

We use several popular datasets for image classifications.

- CIFAR-10 (Krizhevsky & Hinton, 2009): it consists of a training set of $50,000$ $32 \times 32$ color images from 10 classes, and also $10,000$ test images.
- CIFAR-100 (Krizhevsky & Hinton, 2009): it is similar to CIFAR-10 but contains 100 image classes with 600 images for each.
- ImageNet dataset (ILSVRC2012) (Deng et al., 2009): ILSVRC2012 contains 1.28 million training images, and $50k$ validation images over 1000 classes.

## C.2 Parameter Settings

We perform grid searches to choose the best base learning rate for all algorithms over $\{0.0001, 0.001, 0.01, 0.1, 1\}$, the partial adaptive parameter $p$ from $\{1/4, 1/8, 1/16\}$ and the second order momentum parameter $\beta_2$ from $\{0.9, 0.99, 0.999\}$. For SGD with momentum, the base learning rate is set to be $0.1$ with a momentum parameter of $0.9$. For Adam and Amsgrad, we set the base learning rate as $0.001$. For Padam, the base learning rate is set to be $0.1$ and the partially adaptive parameter $p$ is set to be $1/8$ due to its best empirical performance. We also tune the momentum parameters for all the adaptive gradient methods. After tuning, for Adam, Amsgrad, the momentum parameters are set to be $\beta_1 = 0.9$, $\beta_2 = 0.99$. And for Padam, we set $\beta_1 = 0.9$, $\beta_2 = 0.999$. Padam and SGD with momentum use a weight decay factor of $5 \times 10^{-4}$ and for Adam/ Amsgrad we use a weight decay factor of $1 \times 10^{-4}$. For AdamW, the normalized weight decay weight decay factor is set to $2.5 \times 10^{-2}$ for CIFAR and $5 \times 10^{-2}$ for ImageNet. All experiments use cross-entropy as the loss function. The minibatch sizes for CIFAR-10 and CIFAR-100 are set to be $128$ and for ImageNet dataset we set it to be $256$.

## C.3 CNN Architectures

**VGGNet** (Simonyan & Zisserman, 2014): We use a modified VGG-16 architecture for this experiment. The VGG-16 network uses only $3 \times 3$ convolutional layers stacked on top of each other for increasing depth and adopts max pooling to reduce volume size. Finally, two fully-connected layers [6] are followed by a softmax classifier.

**ResNet** (He et al., 2016): Residual Neural Network (ResNet) introduces a novel architecture with "skip connections" and features a heavy use of batch normalization. Such skip connections are also known as gated units or gated recurrent units and have a strong similarity to recent successful elements applied in RNNs. We use ResNet-18 for this experiment, which contains 2 blocks for each type of basic convolutional building blocks in He et al. (2016).

**Wide ResNet** (Zagoruyko & Komodakis, 2016): Wide Residual Network further exploits the "skip connections" used in ResNet and in the meanwhile increases the width of residual networks. In detail, we use the 16 layer Wide ResNet with 4 multipliers (WRN-16-4) in our experiments.

# D Additional Experiments

Figure 4 plots the train loss and test error (top-1 error) against training epochs on the CIFAR-100 dataset. We can see that Padam achieves the best from both worlds by maintaining faster convergence rate while also generalizing as well as SGD with momentum in the end.

Table 2: Test accuracy of VGGNet on CIFAR-100. Bold number indicates the best result.

| Methods | Test Accuracy (%) | | | |
|---|---|---|---|---|
| | 50th Epoch | 100th Epoch | 150th Epoch | 200th Epoch |
| SGD Momentum | $57.37 \pm 0.55$ | $68.63 \pm 0.35$ | $\mathbf{72.89 \pm 0.18}$ | $\mathbf{73.03 \pm 0.13}$ |
| Adam | $62.62 \pm 0.41$ | $66.73 \pm 0.23$ | $66.99 \pm 0.20$ | $67.17 \pm 0.20$ |
| Amsgrad | $63.54 \pm 0.29$ | $66.92 \pm 0.33$ | $67.24 \pm 0.17$ | $67.11 \pm 0.18$ |
| AdamW | $\mathbf{63.88 \pm 0.38}$ | $67.97 \pm 0.20$ | $68.75 \pm 0.19$ | $69.00 \pm 0.10$ |
| Padam | $61.92 \pm 0.22$ | $\mathbf{70.52 \pm 0.30}$ | $72.84 \pm 0.25$ | $72.95 \pm 0.07$ |

Table 2 shows the test accuracy of VGGNet on CIFAR-100 dataset. We can see that for CIFAR-100 dataset, Padam achieves the best test accuracy at 100th, and also keep really close to SGD with momentum at 150th and 200th epochs (differences less than $0.1\%$).

As a side product of our experiments, we also examine the distance that $\boldsymbol{\theta}_t$ has traversed in the parameter space, i.e., $\|\boldsymbol{\theta}_t - \boldsymbol{\theta}_0\|_2$, as shown in many other studies (Hoffer et al., 2017; Xing et al.,

---

[6]For CIFAR experiments, we change the ending two fully-connected layers from 2048 nodes to 512 nodes. For ImageNet experiments, we use batch normalized version (vgg16_bn) provided in Pytorch official package

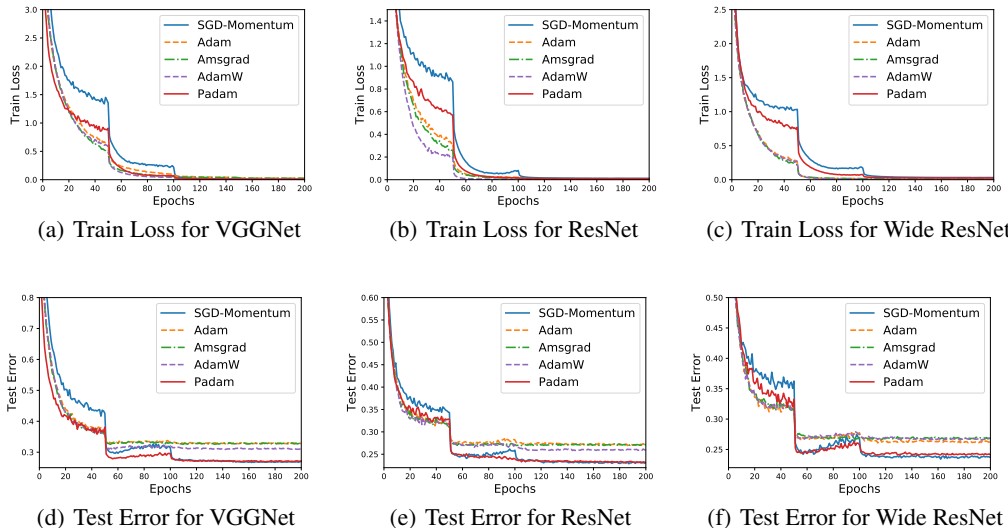

(a) Train Loss for VGGNet     (b) Train Loss for ResNet     (c) Train Loss for Wide ResNet

(d) Test Error for VGGNet     (e) Test Error for ResNet     (f) Test Error for Wide ResNet

Figure 4: Train loss and test error (top-1 error) of three CNN architectures on CIFAR-100. In all cases, Padam achieves the fastest training procedure among all methods and generalizes as well as SGD with momentum.

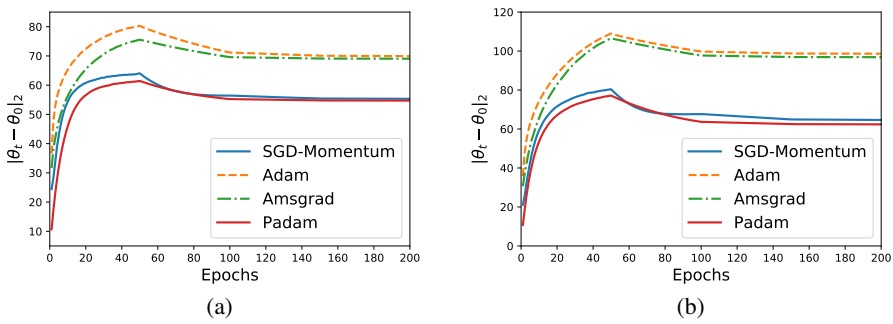

(a)               (b)

Figure 5: Plots for $\|\boldsymbol{\theta}_t - \boldsymbol{\theta}_0\|_2$ against training epochs. Both experiments adopts VGGNet on CIFAR-10 and CIFAR-100 datasets.

2018) discussing the generalization performance of various optimization algorithms for deep learning. Figure 5 shows the plot of the Euclidean distance of weight vector $\boldsymbol{\theta}_t$ from initialization $\boldsymbol{\theta}_0$, against training epochs for ResNet on CIFAR-10 dataset[7]. First, we would like to emphasize that these are the plots only regrading the Euclidean distance, which does not contain any directional information. In other words, the cross in the plot does not mean that the weight vectors in different optimization algorithms actually meet somewhere during the training process. As we can see from the plots, SGD with momentum tends to overshoot a lot at the early stage of the entire training process. This could explain why the convergence of SGD with momentum is slower at early stages. And for Adam/Amsgrad, despite the quick start and less overshooting, the "small learning rate dilemma" (see Section 3) largely limits the weight vector's ability of exploring the parameter space in the middle and later stages, which could explain the bad generalization performance of Adam/Amsgrad. On the other hand, Padam behaves inbetween SGD with momentum and Adam/Amsgrad, with less severe overshooting compared with SGD with momentum, while maintaining a better capability of traversing through the parameter space compared with Adam/Amsgrad. This partly justifies the outstanding generalization performance of Padam.

---

[7]We did not compare with AdamW here since it changes the definition of weight decay and thus have a way larger distance compared with other baselines.

