# OpenReview forum: "Padam: Closing the Generalization Gap of Adaptive Gradient Methods in Training Deep Neural Networks"
_ICLR.cc/2019/Conference_

### Official Review · AnonReviewer1 · 2018-10-19
**Excellent contribution to solving an important problem**

**Rating:** 9
**Confidence:** 3

**Review:**

The authors propose a modification of existing adaptive variants of SGD to avoid problems with generalization. It is known that adaptive gradient algorithms such as Adam tend to find good parameter values more quickly initially, but in the later phases of training they stop making good progress due to necessarily low learning rates so SGD often outperforms them past a certain point. The suggested algorithm Padam achieves the best of both worlds, quick initial improvements and good performance in the later stages.

This is potentially a very significant contribution which could become the next state-of-the-art optimization method for deep learning. The paper is very clear and well-written, providing a good overview of existing approaches and explaining the specific issue it addresses. The authors have included the right amount of equations so that they provide the required details but do not obfuscate the explanations. The experiments consist of a comprehensive evaluation of Padam against the popular alternatives and show clear improvements over them.

I have not evaluated the convergence theorem or its proof since this is not my area of expertise. One thing that stood out to me is that I don't see why theta* should be unique.

Some minor suggestions for improving the paper:

Towards the end of section 2 you mention a non-convergence issue of Adam. It would be useful to add a few sentences to explain exactly what the issue is.

I would suggest moving the details of the grid search for p to the main text since many readers would be interested to know what's typically a good value for this parameter.

Would it make sense to try to adapt the value of p, increasing it as the training progresses? Since that's an obvious extension some comment about it would be useful.

On the bottom of page 6: "Figures 1 plots" -> "Figure 1 plots".

Make sure to protect the proper names in the bibliography so that they are typeset starting with uppercase letters.

---

> ### Author Response · Authors · 2018-11-23
> **Response to Reviewer1**
>
> 1. Thank you for pointing this question. theta^* may not be unique, because even for convex functions, there may exist more than one global minimizer, unless the function is strictly convex. So we change the first equation in Section 4 to be \theta^* \in \argmin_{\theta \in X} \sum_{t=1}^T f_t(\theta), which reflect the fact that theta^* is just one of those global minimizers.   Our proof still holds and it does not affect the convergence result in Theorem 4.2 and Corollary 4.4.
>
> 2. Thank you for your suggestion and we have further mentioned the reason for the non-convergence in Adam in the revision.
>
> 3. Thank for your suggestion and we have moved different p plot to main text in the revision.
>
> 4. Yes, Trying to adapt the value of p could be an interesting idea and future work. Thank you for your suggestion.
>
> 5. Thank you for your suggestion and we have fixed the typos it in the revision.

---

### Official Review · AnonReviewer3 · 2018-10-30
**The contribution is relatively minor and an important baseline is missing in comparison.**

**Rating:** 6
**Confidence:** 4

**Review:**

This paper proposes a small modification to current adaptive gradient methods by introducing a partial adaptive parameter, showing improved generalization performance in several image classification benchmarks.

Pros:
- The modification is simple and easy to implement.
- The proposed method shows improved performance across different datasets, including ImageNet.

Cons:
- Missing an important baseline - AdamW (https://arxiv.org/pdf/1711.05101.pdf) which shows that Adam can generalize as well as SGD and retain faster training. Basically, the poor generalization performance of Adam is due to the incorrect implementation of weight decay.
- The experimental results for Adam are not convincing. It's well-known that Adam is good at training but might perform badly on the test data. However, Adam performs much worse than SGD in terms of training loss in all plots, which is contrary to my expectation. I doubt that Adam is not tuned well. One possible explanation is that the training budget is not enough, first-order methods typically require 200 epochs to converge. So I suggest the authors training the networks longer (make sure the training loss levels off before the first drop of learning rate.).
- Mixing the concept of generalization and test performance. Note that generalization performance typically measures the gap between training and test error. To make the comparison fair, please make sure the training error is zero (I expect both training error and training loss should be close to 0 on CIFAR).
- In terms of optimization (convergence) performance, I cannot think of any reason that the proposed method would outperform Adam (or Amsgrad). The convergence analysis doesn't say anything meaningful.

---

> ### Author Response · Authors · 2018-11-23
> **Response to Reviewer3**
>
> 1. We have added further comparison with AdamW in the revision as you suggested. As you can see from the plots in the revision, AdamW improves the generalization performance comparing with original Adam but there are still generalization gaps left behind, at least in our test settings. In contrast, Padam could achieve basically as good generalization performance as SGD with momentum.
>
> 2. Thank you for your suggestion to train longer. We have followed your suggestion and rerun our CIFAR10/CIFAR100 experiments to 200 epochs as you suggested to make sure each baseline is well optimized. In particular, by this way, we successfully reduced the the training error/loss of Adam/Amsgrad to zero. We hope this would clear your concern.
> For Imagenet experiments, due to the large training test, 100 epochs is more than enough to converge, as can be seen in original papers for VGGNet, ResNet.
>
> Regarding the concept of generalization and test performance, we think generalization error and test error are usually referring to the same thing, while generalization gap is the difference between training error and test error. When we talk about generalization gap, it does not require the training error to be zero. Yet we agree that we should make the training error zero in our experiments, because this is more aligned with the practice of deep learning. Thus we have done that by running the training process for more epochs as you suggested. We hope this addressed your question.
>
> 3.  From the convergence analysis we presented for convex optimization, Padam may not outperform Adam, as you said. Yet the convergence analysis in the paper at least guarantees that Padam’s convergence is also not worse, at least as good as Adam. In fact, in a follow up work (https://arxiv.org/abs/1808.05671), they have proved that the convergence rate of Padam (when 0<p<=1/4) outperforms that Adam for noncovnex optimization. This further backups our experimental findings in this paper.

---

> > ### Comment · AnonReviewer3 · 2018-11-23
> > **Comments**
> >
> > 1 and 3: OK
> >
> > 2: It's reassuring that Adam/Amsgrad are able to reduce training loss to zero. But I hope the authors can use more standard learning rate schedule (e.g., decay the learning rate at epoch 100 and 150.). Empirically, it's better to decay the learning rate after the training error/loss reaches a plateau (50 epoch is far from enough, as shown in figure 2).
> >
> > Regarding the performance of AdamW, I guess the authors didn't tune the weight decay parameter. I think AdamW with carefully chose weight decay factor can match SGD with momentum.
> >
> > Overall, I think the current version deserves to be read since it achieves basically as good generalization performance as SGD with only a simple modification. I increase the score to 6.

---

> > > ### Author Response · Authors · 2018-11-24
> > > **Thank you**
> > >
> > > Thank you very much for your suggestion and increasing your score. We will try using your suggested learning rate schedule. As for weight decay factor for AdamW, we used the parameter suggested in the original paper/github repository. You are right, it could be better since the test environment is not exactly the same. And we will also tune the weight decay factor for AdamW according to your suggestion. As you know, running deep learning experiments at this scale usually takes quite long time. That’s also why we took so long to respond. So we may not be able to include this new experimental results before the end of response period, but we will definitely add these additional experimental results in the camera-ready if needed.

---

### Official Review · AnonReviewer2 · 2018-11-03
**simple generalization of AMSgrad/momentum, good test data/models, results not significant/compelling**

**Rating:** 6
**Confidence:** 4

**Review:**

The idea is simple and promising: generalize AMSgrad and momentum by hyperparameterizing the p=1/2 in denominator of ADAM term to be within [0,1/2], with 0 being momentum case.  It was good to see the experiments use non-MNIST data (e.g. ImageNet, Cifar) and reasonable CNN models (ResNet, VGG).  However, the experimental evaluation is not convincing that this approach will lead to significant improvements in optimizing such modern models in practice.

One key concern and flaw in their experimental work, which was not addressed, nor even raised, by the authors as a potential issue, is that their PADAM approach got one extra hyperparameter (p) to tune its performance in their grid search than the competitor optimizers (ADAM, AMSgrad, momentum).  So, it is not at all surprising that given it has one extra parameter, that there will be a setting for p that turns out to be a bit better than 0 or 1/2 for any given data/model setup and weight initialization/trajectory examined.  So at most this paper represents an existence proof that a value of p other than 0 or 1/2 can be best.  It does not provide any guidance on how to find p in a practical way that would lead to wide adoption of PADAM as a replacement for the established competitor optimizers. As Figures 2 and 3 show, momentum ends up converging to as good a solution as PADAM, and so it doesn't seem to matter in the end that PADAM (or ADAM) might seem to converge a bit faster at the very beginning.

This work might have some value in inspiring follow-on work that could try to make this approach practical, such as adapting p somehow during training to lead to truly significant speedups or better generalization.  But as experimented and reported so far, this paper does not give readers any reason to switch over to this approach, and so the work is very limited in terms of any significance/impact.  Given how simple the modification is, the novelty is also limited, and not sufficient relative to the low significance.

---

> ### Author Response · Authors · 2018-11-23
> **Response to Reviewer 2**
>
> 1. You are right we indeed need one extra hyperparameter p to achieve the best performance. Yet we think it is more than just another “hyperparameter”. Our work has a clear and strong motivation, as we described in the introduction, which is solving the very important generalization gap problem of adaptive gradient methods such as Adam. This is a long standing problem, which prevents the good performance of Adam for training morden deep neural networks. Our solution proposed in this paper has well solved this problem.
>
> In terms of how to choose p, we recommend doing binary search over the grid {1/4, 1/8,  1/16}.  We have added this suggestion in the revision.
>
> As for the cost of tuning one extra parameter, we do not think there will be much trouble. If you think about Adam, which also introduce a new set of parameters (beta1,beta2) compared with its predecessors. Yet people did not complain this for Adam because the choice of these parameters tends to be quite stable and simple. The same applied here, in general case, 1/8 is a quite stable and effective choice of p as we have tested base on extensive experiments. Therefore the tuning process of p does not cost much.
>
> 2. As you said, converging faster is just a plus, the key point is that we fix the the generalization gap of adaptive gradient methods for morden deep neural networks. This means a lot, because for small-scale neural networks, people would like to use adaptive gradient methods such as Adam for fast convergence. Now for large-scale deep neural networks, people now can also use adaptive gradient methods (Padam) since the generalization gap issue has been fixed. We believe our contribution in this paper is significant, especially for practitioners who used Adam a lot for small-scale neural networks, but had to give up using Adam for training  morden large-scale neural networks, because of its unappealing generalization gap.
>
> 3. We think it is hard to judge the contribution/novelty of one work significant or not based on whether the modification is simple or not. In some sense, Adam is also a simple modification given RMSprop and Adagrad. Yet Adam still stands out given its empirical performances. In this work, we also try to deliver a useful, practical algorithm that could fix a key issue (generalization gap) in existing adaptive gradient methods and therefore, save practitioners from the difficult choice of using SGD or Adam (Now they can all use Padam).

---

### Meta-Review · Area_Chair1 · 2018-12-13

**Recommendation:** Reject
**Confidence:** 4

**Metareview:**

This paper proposes a simple modification of the Adam optimizer, introducing a hyper-parameter 'p' (with value in the range [0,1/2]) parameterizing the parameter update:
theta_new = theta_old + m/v^p
where p=1/2 falls back to the standard Adam/Amsgrad optimizer, and p=0 falls back to a variant of SGD with momentum.

The authors motivate the method by pointing out that:
 - Through the value of 'p', one can interpolate between SGD with momentum and Adam/Amsgrad. By choosing a value of 'p' smaller than 0.5, one can therefore use perform optimization that is 'partially adaptive'.
 - The method shows good empirical performance.

The paper contains an inaccuracy, which we hope will be solved before the final version. The authors argue that the 1/sqrt(v) term in Adam results in a lower learning rate, and the authors argue that the effective learning rate "easily explodes" (section 3) because of this term, and that a "more aggressive" learning rate is more appropriate. This last point is false; the value of 1/sqrt(v) can be smaller or larger than 1 depending on the value of 'v', and that a decrease in value of 'p' can result in either an increase or decrease in effective learning rate, depending on the value of v. The value of 'v' is a function of the scale of loss function, which can really be arbitrary. (In case of very high-dimensional predictions, for example, the scale of the loss function is often proportional with the dimensionality of variable to be modeled, which can be arbitrarily large, e.g. in image or video modeling the loss function tends to be of a much larger scale than with classification.)

The authors promise to include a comparison to AdamW [Loshchilov, 2017] that includes tuning of the weight decay parameter. The lack of this experiments makes it more difficult to make a conclusion regarding the performance relative to AdamW. However, the methods offer potentially orthogonal (and combinable) advantages.

[Loshchilov, 2017] https://arxiv.org/pdf/1711.05101.pdf